



# De Long Trough: A newly discovered glacial trough on the East Siberian Continental Margin

Matt O'Regan[1,2], Jan Backman[1,2], Natalia Barrientos[1,2], Thomas M. Cronin[3], Laura Gemery[3], Nina Kirchner[2,4], Larry A. Mayer[5], Johan Nilsson[2,6], Riko Noormets[7], Christof Pearce[1,2,8], Igor Semilietov[9,10], Christian Stranne[1,2,5], Martin Jakobsson[1,2].

[1] *Department of Geological Sciences, Stockholm University, Stockholm, 106 91, Sweden*

[2] *Bolin Centre for Climate Research, Stockholm University, Stockholm, Sweden*

[3] *US Geological Survey MS926A, Reston, Virginia, 20192, USA*

[4] *Department of Physical Geography (NG), Stockholm University, SE-106 91 Stockholm, Sweden*

[5] *Center for Coastal and Ocean Mapping, University of New Hampshire, New Hampshire 03824, USA*

[6] *Department of Meteorology, Stockholm University, Stockholm, 106 91, Sweden*

[7] *University Centre in Svalbard (UNIS), P O Box 156, N-9171 Longyearbyen, Svalbard*

[8] *Department of Geoscience, Aarhus University, Aarhus, 8000, Denmark*

[9] *Pacific Oceanological Institute, Far Eastern Branch of the Russian Academy of Sciences, 690041 Vladivostok, Russia*

[10] *Tomsk National Research Polytechnic University, Tomsk, Russia*

*Correspondence to*: Matt O'Regan (matt.oregan@geo.su.se)

**Abstract.** Ice sheets extending over parts of the East Siberian continental shelf have been proposed during the last glacial period, and during the larger Pleistocene glaciations. The sparse data available over this sector of the Arctic Ocean has left the timing, extent and even existence of these ice sheets largely unresolved. Here we present new geophysical mapping and sediment coring data from the East Siberian shelf and slope collected during the 2014 SWERUS-C3 expedition (SWERUS-C3: Swedish – Russian – US Arctic Ocean Investigation of Climate-Cryosphere-Carbon Interactions). The multibeam bathymetry and chirp sub-bottom profiles reveal a set of glacial landforms that include grounding zone formations along the outer continental shelf, seaward of which lies a >65 m thick sequence of glaciogenic debris flows. The glacial landforms are interpreted to lie at the seaward end of a glacial trough – the first to be reported on the East Siberian margin, here referred to as the De Long Trough because of its location due north of the De Long Islands. Stratigraphy and dating of sediment cores show that a drape of acoustically laminated sediments covering the glacial deposits is older than ~50




cal. kyr BP. This provides direct evidence for extensive glacial activity on the Siberian shelf that pre-dates the Last Glacial Maximum and most likely occurred during the Saalian (Marine Isotope Stage [MIS] 6).

## 1 Introduction

The glacial history of the Siberian continental shelf of the East Siberian Sea is poorly known and marine geological and geophysical data from this region are scarce. Most of the area is shallower than 120 m implying that it was exposed during the sea-level lowstand of the Last Glacial Maximum (LGM) and the larger glaciations following the mid-Pleistocene transition (Lambeck et al., 2014; Rohling et al., 2014), even considering glacial isostatic adjustments (Klemann et al., 2015) (Fig. 1). One consequence

of the shallowness of the East Siberian shelf is that submarine glacial landforms, signifying the presence of an ice sheet (Dowdeswell et al., 2016), may have been eroded during regressive and transgressive cycles.

On formerly glaciated margins, areas of fast-streaming glacial ice are recognised by the presence of

glacially excavated cross-shelf troughs (CSTs) (Batchelor and Dowdeswell, 2014). CSTs and their sedimentary archives are extensively used to identify the existence, extent and dynamics of former ice sheets (Polyak et al., 1997; Anderson et al., 2002; Winsborrow, et al., 2010; Kirshner et al., 2012; Hogan et al., 2010, 2016). Twenty glacially excavated troughs emptying directly into the Arctic Ocean are identified in existing bathymetric and seismic data from north of Fram Strait (Batchelor and

Dowdeswell, 2014) (Fig. 1). Several of these can be traced back into tributary fjords on adjacent landmasses, or towards the center of former ice sheets, and are particularly pronounced along the Barents-Kara and North American margins (Batchelor and Dowdeswell, 2014; Jakobsson, 2016) (Fig. 1). By contrast, the shallow shelves of the East Siberian and Chukchi Seas lack any identified CSTs. Despite the absence of these diagnostic features, ice sheets extending over parts of the East Siberian

continental shelf have been proposed in literature during the Last Glacial Maximum (LGM) (Toll, 1887; Hughes et al., 1977; Grosswald, 1990), MIS 6 (Basilyan et al., 2008, 2010; Jakobsson et al., 2016) and





the larger Pleistocene glaciations that followed the mid-Pleistocene transition (Colleoni et al., 2016;
Niessen et al., 2013).

The existence of an ice sheet on the New Siberian Islands was first proposed by Toll (1887) based on
the widespread occurrence of ice wedges, which he interpreted as relict glacial ice. Although ice-
wedges are today known to be formed in permafrost by refreezing of water flowing into cracks, glacio-
tectonised Cretaceous and Cenozoic sediments on the New Siberian Islands do contain thick inclusions
of ice interpreted to originate from an ice sheet, and are overlain by conformable Quaternary sediments
(Basilyan et al., 2008; 2010). The orientation of the glacio-tectonic features indicates that glacial ice on
the New Siberian Islands flowed from a north-northeast direction, and likely nucleated over the De
Long Islands, where small glaciers remain today (Basilyan et al., 2008) (Fig. 1). Uranium-thorium
($^{230}$Th/$^{234}$U) dating on mollusc shells in sediments overlying the glacial deposits made Basilyan et al.
(2010) conclude that the glaciation may have been centred around 135 ka, during MIS 6.

Another line of evidence for glacial ice on the Siberian continental shelf is the presence and orientation
of glaciogenic features and sedimentary deposits mapped on the seafloor in the adjacent Arctic Basin.
These glacial features are mapped on the lower slope of the East Siberian Sea, and on the crest of
shallower ridges and plateaus of the Arctic Ocean (Niessen et al., 2013; Jakobsson et al., 2016).
Streamlined glacial lineations on the seabed of the Arlis Plateau and the base of the East Siberian
continental slope, have orientations that indicate ice flow from the East Siberian shelf (Niessen et al.,
2013; Jakobsson et al., 2016) (Fig. 1). Niessen et al. (2013) speculate that the modern water depths of
these features, ranging between about 900 and 1200 meters below sea level (mbsl), imply an ice
thickness on the East Siberian continental shelf of up to 2 km. Glacial lineations also exist on a heavily
ice-scoured crest of the Southern Lomonosov Ridge (81° N 143° E), where, together with a gently
sloping stoss side towards the Makarov Basin and a steep lee side facing the Amundsen Basin, clearly
indicate grounded glacial ice flowing from the East Siberian shelf (Jakobsson et al., 2016) (Fig. 1).
Combined with the orientations of glacial features on the Chukchi Borderland (Dove et al., 2014),
Alaskan Beaufort Slope (Engels et al., 2008), and central Lomonosov Ridge (Jakobsson et al., 2010,

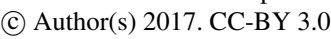


2016), it is suggested that large ice shelves in the Amerasian Arctic existed during past glacial periods, and were fed from ice discharging from the East Siberian shelf and North America (Jakobsson et al., 2010). Recent mapping on the Lomonosov Ridge led Jakobsson et al. (2016) to propose that an ice shelf was not limited to the Amerasian Arctic Ocean and instead covered the entire central Arctic Ocean.

5 Marine sediments atop the mapped glacial features in the central Arctic Ocean have consistently been dated to MIS 5.5, implying that the large central Arctic Ocean ice shelf existed during MIS 6 (Jakobsson et al., 2010, 2016). However, marine based glaciers large enough to ground on the Chukchi Borderland may have existed after MIS 5.5 (Polyak et al., 2007), feeding a thinner ice shelf that covered parts of the Western and central Arctic (Jakobsson et al., 2014).

Despite the mounting evidence for glacial ice on the East Siberian shelf, our ability to define its extent and timing remains limited. In part this is due to the lack of glacial morphology on the shelf that could be used to link the terrestrial observations with marine mapping results in deeper water settings of the Arctic Ocean. However, the sparse data availability across the East Siberian shelf implies that the

15 absence of known submarine geomorphological features does not preclude their existence. Here we present new geophysical and sedimentological evidence for a glacial trough north of the De Long and New Siberian Islands on the outer margin of the East Siberian shelf. The trough was most likely occupied by glacial ice during MIS 6, and certainly free from glacial ice during the LGM.

## 2 Methods

20 ### 2.1 Expedition

The data presented in this paper were acquired on Leg 2 of the SWERUS-C3 2014 Expedition on IB *Oden*, which departed August 21 from Barrow, Alaska, and ended October 3 in Tromsø, Norway (Fig 2). The data include multibeam bathymetry, chirp sub-bottom profiles and analyses of sediment cores collected along a 225-km long downslope transect spanning water depths of 115 to 1800 mbsl (Fig. 3).



## 2.2 Geophysical mapping

A brief summary of the geophysical mapping methods during the SWERUS-C3 expedition is included here, further details are described in Jakobsson et al. (2016). Multibeam bathymetry and sub-bottom profiles were collected with the Kongsberg EM 122 (12 kHz, 1°x1°) multibeam and integrated SBP 120 (2-7 kHz, 3°x3°) chirp sonar installed in IB *Oden*. This system has a Seatex Seapath 330 unit for integration of GPS navigation, heading and attitude. Temperature and salinity data from CTD (Conductivity, Temperature, Depth) stations and regular XBT (Expendable Bathy Thermograph) casts were used to calculate sound speed profiles for calibration of the multibeam. Multibeam bathymetry was post-processed using a combination of the Caris and Fledermaus-QPS software. Sub-bottom profiles were acquired using a 2.5-7 kHz chirp pulse. The chirp sonar profiles were post-processed and interpreted using a combination of the open source software OpendTect created by dGB Earth Sciences and tools provided by the Geological Survey of Canada (Courtesy Bob Courtney).

## 2.3 Sediment cores

Four sediment cores (inner diameter of 100 mm) are presented in this study (Table 1). They were collected using either a piston (PC) or gravity (GC) corer, both rigged with a 1360-kg core head. The unsplit sediment cores were allowed to equilibrate to room temperature ($20^{\circ}$C) and logged shipboard on a Multi-Sensor Core Logger (MSCL). Bulk density, compressional wave velocity (p-wave) and magnetic susceptibility (Bartington loop sensor) were measured at a downcore resolution of 1 cm. The cores were split and described shipboard, and imaged using a digital line-scanning camera. The undrained shear strength ($S_U$) of the sediments was measured using a CONTROLS-group liquid limit penetrometer (fall cone). The fall cone test was performed according to ISO-TS-17892-6 (Swedish standards institute) at a downcore resolution of approximately 30 cm. For most measurements a 60°/82g cone was used, but it some instances a heavier weight (60°/112g) or narrower cone (30°/62g) was used to achieve the recommended cone penetration depth of 4-20 mm.

The undrained shear strength was calculated using the cone geometry, weight and penetration,

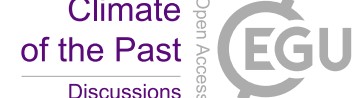



$$S_U = Kg \frac{Q}{h^2}$$

where $K$ is a cone dependent constant (0.8 for the $30^o$ cone and 0.27 for the $60^o$ cone), $Q$ is the cone weight (g), $g$ the acceleration due to gravity (9.81 m/s$^2$) and $h$ the cone penetration (mm).

Shorebased measurements on the split cores were conducted at the Department of Geological Sciences,
Stockholm University. These included additional magnetic susceptibility measurements, grain size and XRF-core scanning. The magnetic susceptibility was re-measured on the MSCL using a Bartington point sensor. Compared to the loop sensor measurements on the whole core, the point sensor provides superior horizontal resolution (lower effective sensor length) but only measures the susceptibility of sediments in the upper few millimeters from the split core surface.

Sediment grain size (2 µm to 2 mm) was measured at a 5 cm downcore resolution using a Malvern Mastersizer 3000 laser diffraction particle size analyzer. Wet samples were immersed in a dispersing agent (<10% sodiumhexametaphosphate solution) and placed in an ultrasonic bath to ensure full particle disaggregation before analyses. The mean grain size and sorting were calculated using the Geometric
method of moments (Blott and Pye, 2001).

Elemental abundances were measured on the archive half of the split cores using an ITRAX XRF core scanner. Analyses were made with a Mo tube set at 55 kV and 50 mA, a step size of 2 mm and a counting time of 20 s. The data were normalized by the incoherent + coherent scattering and the ratio of
Ca/Ti used to help stratigraphically correlate the sediment cores.

**2.4 Dating**

Accelerator mass spectrometry (AMS) radiocarbon measurements were made on samples containing the planktonic foraminifer *Neogloboquadrina pachyderma*, mixed benthic foraminifera or mollusk shells.
These were performed at either the National Ocean Sciences Accelerator Mass Spectrometry (NOSAMS) facility at Woods Hole Oceanographic Institution, Massachusetts, or the Lund University





Radiocarbon Dating Laboratory (Lu), Sweden. Six radiocarbon ages were obtained from the core catcher of 20-GC1, and single range finding ages obtained from both 23-GC1 and 24-GC1 (Table 2).

## 3 Results

### 3.1 Sub-bottom stratigraphy

The sub-bottom stratigraphy from the outer shelf 20-GC1 (77° 21.5' N, 163° 2.0' E) to the shelf break is divided into six acoustic units (Fig. 4). Unit 1 is a thin, discontinuous, largely incoherent veneer of sediments with a sharp basal contact (R1) that can be traced into water depths of between 260-300 mbsl. Above 260 mbsl, it is underlain by Unit 2, mostly composed of horizontally layered or dipping and truncated reflection packages interspersed with intervals of acoustically transparent material (Fig 4).

The base of Unit 2 is never imaged in the sub-bottom data, but on the shallowest regions of the survey area, it extends more than 50-60 meters below the seafloor (mbsf). Between 260 and 300 mbsl, Unit 1 transitions into a coherent and laterally continuous acoustically layered sequence (Unit 3) (Fig. 4). The R1 reflector is no longer distinguishable.

The base of Unit 3 is defined by a hummocky reflector (R2) that overlies two acoustically transparent facies (Units 4 and 5). The thickness of Unit 4 varies considerably along the track line, in places infilling v-shaped wedges, and near the shelf break, thickening in two prominent sedimentary deposits, here referred as mounds M1 and M2 (Figs. 4, 5). Although the exact lengths cannot be determined because of the orientation of the ship track, M1 is ~10-15 km long, and at its maximum height

approaches 30 ms TWT (24-27 m using a p-wave velocity of 1600-1800 m/s). M2 is smaller, ~5-10 km long, but attains a similar thickness to M1.

Although similar in appearance, Units 4 and 5 are separated by an often strong and planar reflector (R3), which exists seaward of 360 mbsl (480 ms TWT). A third sedimentary deposition, mound M3, is

recognised within Unit 4 and lies landward of M1 and M2 within seismic Unit 5 (Figs. 4, 5). It is ~10-15 km long, and has a thickness of 35 ms TWTW (28-32 m using a p-wave velocity of 1600-1800 m/s).





Superposition of acoustic units interpreted from the sub-bottom data indicates that M3 is the oldest of these features (Figs. 4, 5).

At the shelf break, the base of Unit 4 and 5 are separated by a distinct acoustically transparent, wedge-shaped sediment package (Unit 6) whose upper boundary is defined by R3 and lower boundary by R5 (Figs. 4, 5). Seaward of the shelf break, only the uppermost acoustically laminated unit (Unit 3) can be laterally traced (Fig. 5). It overlies a complex sequence of strong undulating, discontinuous acoustically laminated sections, interspersed with thicker acoustically transparent intervals (Fig. 6). This sequence extends to the maximum depth of the surveyed area (>2000 mbsl).

## 3.2 Multibeam Bathymetry

There are no prominent morphological seafloor features distinguishable along the transect where only one single swath of multibeam bathymetry was collected (Fig. 7). Where the small survey was made in the vicinity of coring site 23-GC1, the three sedimentary mounds identified in the sub-bottom profiles (Fig. 4) are visible as morphological seafloor expressions (Fig. 7). The multibeam bathymetry reveal that the mounds extend laterally as far as the survey covers. Furthermore, it is seen in the bathymetry that the sub-bottom profiles presented in Figure 4 crosses M1 and M2 at an angle of about 45°, while the ship changed course when crossing M3 to follow its direction for ~3.5 km before returning to the old course. This implies that the expression of M3 likely appears more vertically subdued and laterally extensive than it is.

The bathymetry from the International Bathymetric Chart of the Arctic Ocean (IBCAO) (Jakobsson et al., 2012) is shown as a backdrop in Figure 7. The depth difference between the newly collected multibeam bathymetry with IB *Oden* and IBCAO is largest, reaching >100 m, in the area of the shelf break where M1-M3 are located.



### 3.3 Sediment stratigraphy and chronology

Three of the sediment cores presented in this study penetrated to the base of acoustic Unit 3 (22-PC1, 23-GC1 and 24-GC1), while the shallowest core (20-GC1) sampled sediments from acoustic Unit 1. The base of 22-PC1, 23GC-1 and 24GC-1, all contained a dark grey, poorly sorted sequence of coarser

grained sediments (sedimentary unit B) (Fig. 8). The transition into this lower sedimentary unit is abrupt, and is reflected by a substantial increase in sediment bulk density, compressional-wave velocity and magnetic susceptibility (Fig. 8). The undrained shear strength increases in Unit B, but remains relatively low, only exceeding 15 kPa in 23-GC1 (Fig. 8). Point and loop sensor susceptibility measurements coincide throughout sedimentary Unit A, and diverge in the coarser grained Unit B (Fig.

8). This likely reflects the presence of larger magnetic clasts irregularly distributed within the coarse grained sediments. In the upper 50 cm of 22-PC1, 23-GC1 and 24-GC1, a less pronounced coarser grained interval is present, again displaying higher bulk density, but with no notable change in the magnetic susceptibility. Correlation between the two sedimentary units (A and B) in 22-PC1, 23-GC1 and 24-GC1 is straightforward using the grain size and physical property data (Fig. 9). It is further

refined by incorporating the Ca/Ti ratio from the XRF-scanning data (Fig. 9).

Core 20-GC1, obtained from the shallow shelf, only recovered sediments from acoustic Unit 1. It also contained a coarser grained, dark grey facies at its base, transitioning into lower density and susceptibility sediments above 0.5 mbsf. Six radiocarbon dates were obtained from below 0.55 mbsf in

this core, and indicate that the basal sequence is younger than 14 cal kyr BP, and deposited after sea level transgression of this site following the last glacial cycle (Cronin et al., this volume). Although the base of the core has similar sedimentary characteristics to Unit B in 22-PC1, 23-GC1 and 24-GC1, it is not syndepositional. Radiocarbon dates from a mollusk shell in 23-GC1 (1.69-1.86 mbsf) and a foraminiferal bearing interval in 24-GC1 (1.91-1.93 mbsf) return ages of 33200±560 and 43000±1800

$^{14}$C yrs BP respectively. Calibrated median ages, neglecting any additional ΔR, and reported to 2-sigma are 37000 $\pm^{2600}_{1300}$ and 46300 $\pm^{3500}_{2600}$ cal. yrs. BP (Table 2). The radiocarbon dates are consistent with the stratigraphic correlation between the cores (Fig. 9). Investigations into the occurrence of calcareous nannofossils (performed every 10 cm) revealed a single *Emiliania huxleyi* at 2.28 mbsf in 23-GC1,



which indicates that the sediments at this level are younger than MIS 6 (Backman et al., 2009). The basal sediments in 20-GC1, likely correlate to the uppermost section of coarser sediments in 22-PC1, 23-GC1 and 24-GC1.

## 4 Discussion

### 4.1 The De Long Trough

Cross-shelf troughs (CSTs) are areas formerly occupied by fast-streaming glacial ice and are diagnostic features for the presence of former ice sheets (Dowdeswell et al., 2016). They terminate in a calving front where large volumes of icebergs are discharged into the ocean or feed into ice shelves. Within a CST, large asymmetrical sedimentary wedges mark still-stands in the streaming-ice, and are known as grounding zone wedges (GZWs). They commonly occur near the shelf break, and are oriented transverse to the ice-flow direction (Batchelor and Dowdeswell, 2015). Seaward of the shelf break, large volumes of subglacial sediments are discharged onto the slope and form bathymetrically prominent Trough Mouth Fans (TMFs) (Ó Cofaigh et al., 2003). These are composed of stacked glaciogenic debris flows, deposited while the ice was at or near the shelf break and are interbedded with ice-distal or open-water marine sediments.

The geophysical data collected from the outer shelf and slope of the East Siberian Sea, north of the De Long Islands, are sparse but contain evidence for many elements commonly associated with a CST. These are described below, and provide the first evidence for a CST on the Siberian shelf, that we hereafter refer to as the *De Long Trough*. The dimensions of the De Long Trough and associated glaciogenic features are compared to those from other high-latitude glacial troughs recently compiled by Batchelor and Dowdeswell (2014, 2015).

### 4.1.1 Grounding zone deposits

The sedimentary mounds (M1, M3) recognised in the sub-bottom data (Figs. 4,5) are interpreted as grounding zone deposits, and closely resemble GZWs. This inference is based on their position close to the shelf break, geometry and acoustic stratigraphy. The outmost sedimentary mound (M2), which is



mapped in more detail, displays pronounced lateral variations in thickness (Fig. 7) and more closely resembles a series of terminal moraines (Batchelor and Dowdeswell, 2015). The limited mapping of these features makes the absolute discrimination between M1, M2 and M3 uncertain, as is their interpretation as GZWs and terminal moraines respectively (Fig. 7). As both types of features are found

at the terminus of marine based ice streams (Dowdeswell et al., 2016), either interpretation supports the more general conclusion that an ice stream existed on the outer East Siberian continental shelf. The absence of an acoustic reflector separating M1 and M2 suggests that they were formed in close succession, during ice stream retreat. Usually a GZW is interpreted as a sign of that the ice stream continued in a floating ice shelf from the grounding line (Batchelor and Dowdeswell, 2015).

The dimensions of the interpreted GZWs (10-15 km long and 24-32 m high) are comparable to the majority of high-latitude GZWs, which tend to be less than 15 km long and 15-100 m thick (Batchelor and Dowdeswell, 2015). The features in De Long Trough compare with smaller GZWs found in the Northern and Western Barents Sea, Northwestern Greenland, Antarctica and the Mackenzie Trough in

the Canadian Beaufort Sea (Batchelor and Dowdeswell, 2015).

### 4.1.2 Cross-shelf trough

The location of the grounding line deposits within a broad bathymetric depression that ends at the shelf break, suggests that they are features preserved with a glacially excavated trough. This trough can be

identified in the Bathymetric Chart of the Arctic Ocean (IBCAO) (Jakobsson et al., 2012) (Fig. 3). However, bathymetric mapping during SWERUS-C3 reveals substantial differences in water depth compared to IBCAO (Fig. 7a). Results from mapping indicate a lower gradient along the base of the trough between 400-500 mbsl where the grounding line deposits are mapped, and a steeper slope beyond them, between 500 and 1000 mbsl. Dimensions of De Long Trough at the shelf break, derived

from IBCAO, are between 40-70 km wide, with a depth of 140 m (Fig. 3). The true depth at the shelf break is closer to 100 m given the new mapping data (Fig. 7a).



The dimensions of 75 Arctic CSTs, reviewed by Batchelor and Dowdeswell (2014), have modal lengths of 150-200 km (with more than 50% of them being between 50 and 200 km), widths of 20-40 km, and depths of 300-400 m. Batchelor and Dowdeswell (2014) suggest that trough width and length are controlled by the volume of ice and sediment flowing through it, which depends on drainage basin size

and the duration/number of times that a paleo-ice stream was active. Conversely, trough depths are more variable, and probably controlled by the number of past glaciations, the underlying geology and tectonic setting. Although the length of the interpreted trough is not accurately defined by the new mapping data, the distance between 22-PC1 and 23-GC1 is 105 km, and provides a minimum estimate for its length. Its width (40-70 km) falls within the modal range of other Arctic troughs, while its depth

(100-140 m) is substantially shallower than any recognised Arctic CSTs.

### 4.1.3 Trough mouth fan

The acoustic stratigraphy and morphology of the continental slope seaward of the GZWs is typical of Trough Mouth Fan (TMF) deposits described on other high-latitude continental margins (Ó Cofaigh et

al., 2003; Dowdeswell et al., 2016; Batchelor and Dowdeswell, 2014). The stacked acoustically transparent units (Fig. 6) are interpreted as glaciogenic debris flows, composed of sediments delivered to the shelf edge by the ice stream (Elverhøi et al. 1997; Laberg and Vorren, 1995; Taylor et al., 2002). Laterally discontinuous lenses of acoustically transparent sediment are interpreted as down-slope deposits, originating from mass wasting on the upper slope. The occurrence of acoustically laminated

intervals within the sediment may be indicative of periods of ice-distal sedimentation, when the ice stream retreated from the shelf break.

Trough mouth fans can usually be identified in generalised bathymetric charts by a bulge in bathymetric contours along the continental slope, and a similar morphology is evident in front of De Long Trough

(Fig. 3). TMF's commonly approach areas of $10^3$-$10^5$ km$^2$ and are most pronounced on shallow continental slopes (<4°) (Ó Cofaigh et al., 2003; Dowdeswell et al., 2016;). The TMF in front of De Long Trough has an average slope angle of 1.2° and is steeper above 1300 mbsl (1.6°) than below





(0.95°). The area interpreted as a TMF (Fig. 3) totals 6540 km$^2$. Sub-bottom data does not penetrate to the base of the glaciogenic debris flow sequence, which is greater than 65 m thick (90 ms TWT).

**4.2 Timing and association with ice sheets on the Siberian Shelf**

The radiocarbon date from 24-GC indicates that the last episode of glacial activity in De Long Trough occurred before $46300 \pm_{2600}^{3500}$ cal. yrs BP (Table 2). This is supported by the date in 23-GC1, and the occurrence of *E. huxleyi* at 2.28 mbsf in 23-GC1, indicating that sediments overlying the glacial features are younger than MIS 6 (~130 ka). This implies that glacial ice did not occupy De Long Trough

during peak global ice volumes (MIS 2, 14-29 ka) of the last glacial cycle (MIS 2-4, 14-71 ka). The age constraints from this study place the occupation of glacial ice in the trough either during MIS 4 (57-71 ka), a stadial during MIS 5, or during the penultimate glaciation (MIS 6, 130-191 ka).

The absence of glacial ice during the LGM is consistent with the existence of permafrost across much of

the submarine East Siberian shelf (Romanovskii et al., 2004; Nicolsky et al., 2012), the reported absence of ice on Wrangel island during the LGM (Gaultieri et al., 2005), and the comparatively limited extent of the Kara ice sheet on the Barents Sea (Möller et al., 2015). The lack of glacial ice in the East Siberian Sea and the Kara Sea during the LGM are both ascribed to generally arid conditions due to a reduction in atmospheric moisture supply to these regions (Gaultieri et al. 2005; Möller et al., 2015).

Glacial landforms indicating ice flow from the East Siberian shelf that are mapped on the continental slope of the East Siberian Sea, Arlis Plateau and southern Lomonsov Ridge, were also formed prior to the LGM (Niessen et al., 2013; Jakobsson et al., 2016), and are consistent with the absence of glacial ice in the De Long Trough during this time.

The De Long Trough notably connects to the reconstructed ice extents around the De Long and New Siberian Islands (Basilyan et al., 2008, 2010) (Fig. 1). Therefore, it is reasonable to assume that glacial activity in the trough is associated with known glaciations of the New Siberian Islands. Uranium-thorium ($^{230}$Th/$^{234}$U) dating of mollusc shells from undeformed Quaternary marine sediments overlying relict glacial sheet ice on the New Siberian Islands implies that glaciation was older than 84.7 (–



6.2/+6.6) ka (Basilyan et al., 2008, 2010). This age is supported by radiocarbon dates from mammal bones obtained from continental sediments above the marine deposits, which returned an oldest date of 48.6 ± 1.5 $^{14}$C kyrs (Basilyan et al., 2008, 2010). These results suggest that glacial ice was not present on this region of the shelf during MIS 4 (57-71 ka). In fact, considering the influence of rejuvenation of

uranium by groundwater flow, Basilyan et al. (2008, 2010) argue that the true age of the marine molluscs is closer to 135 ka, coinciding with the end of MIS 6. An MIS 6 icestream occupying De Long Trough is consistent with age constraints provided in this study, and would fit into the larger picture of an extensive Arctic ice shelf that was fed, in part, by glacial ice on the East Siberian shelf (Jakobsson et al., 2016).

The current data does not allow us to dismiss the possibility that a smaller local ice cap developed over the De Long Islands and fed an ice stream in De Long Trough during a stadial of MIS 5. Late Quaternary glacial extents in the Kara Sea, specifically over the Taimyr Peninsula and the Severnaya Zemlya archipelago, were considerably larger during MIS 5d (109 ka) and 5b (87 ka), compared to the

MIS 4 and LGM extents (Möller et al., 2015). However, when considering the most recent estimates for the age of glacial ice on the New Siberian islands (Basilyan et al., 2008, 2010), and the date of deep-water glacial features that indicate ice flow directions from the East Siberian shelf (Jakobsson et al., 2016), the most plausible explanation is that an ice stream was active in the De Long Trough during MIS 6. Additional research needs to focus on 1), establishing the connection between glacial ice in De

Long Trough and the existence of a larger ice sheet that covered much of the East Siberian shelf (Fig. 1) and 2), acquiring more detailed dating of the sedimentary sequences overlying the glacial deposits to determine if ice re-occupied the trough during a stadial of MIS 5.

### 4.3 Sea level variations and sedimentation on the continental slope

One of the remarkable observations based on the stratigraphy of sediment cores 22-PC1, 23-GC1 and 24-GC1, is that there does not appear to be a dramatic increase in sediment delivery to the outer shelf and slope during the last glacial cycle. This is despite fluctuating eustatic levels that would have seen the repeated exposure and flooding of the shelf. This is at odds with observations in the river-dominated





Laptev Sea (Bauch et al., 2001), and the generally inferred influx of sediments to the outer shelf and slope during periods of transgression (Wegener et al., 2015). Reworking of sediments above ~260 mbsl is evidenced in the sub-bottom data (Acoustic Unit 1 sediments being reworked by sea-level transgression) (Fig. 4), but little to no influence is seen in deeper sections. Therefore, the acoustically

transparent intervals found on the continental slope in front of De Long Trough (Fig. 6), and described further east in Parasound data from the East Siberian Slope (Niessen et al., 2013), are not a consequence of eustatic sea level variations, but originate from glacial activity on the shelf.

## 5. Conclusions

Geophysical and sediment coring data collected on Leg 2 of the 2014 SWERUS-C3 expedition reveals a set of grounding zone deposits at the shelf break of the East Siberian Sea that lie within a distinct bathymetric depression interpreted as a glacial trough. This provides the first evidence for a glacial trough on the East Siberian continental shelf and direct evidence for an ice sheet on the Siberian shelf. The dimensions of the grounding zone deposits and glacial trough conform to the dimensions on many

of the smaller, previously recognized cross shelf troughs in the Arctic. A thick sequence of glaciogenic debris flows exist seaward of the grounding zone deposits and form a notable trough mouth fan. The ice stream occupying the trough was likely connected to glacial ice over the De Long and New Siberian Islands. Multiple lines of evidence indicate that the trough was occupied by an ice stream during the penultimate glaciation (MIS 6).

**Acknowledgement**

We thank the supporting crew and Captain of I/B *Oden* and the support of the Swedish Polar Research Secretariat. Many thanks to Carina Johansson of the Department of Geological Sciences, Stockholm University for laboratory assistance. This research and expedition was supported by the Knut and Alice Wallenberg Foundation (KAW). Individual researchers received support from the Swedish Research Council

(Jakobsson/Coxall; 2012-1680, O'Regan; 2012-3091, Stranne; 2014-478), the U.S. National Science Foundation (Mayer; PLR-1417789), US Geological Survey Climate & Land Use R&D Program (Cronin and Gemery), and Russian Government (Semiletov: grant no. 14.Z50.31.0012). Pearce received funding from the Danish Council



for Independent Research, Natural Sciences (DFF-4002-00098_FNU). Any use of trade, firm, or product names is for descriptive purposes only and does not imply endorsement by the U.S. Government. Data shown in the article acquired during the SWERUS-C3 expedition in 2014 are available through the Bolin Centre for Climate Research database: http://bolin.su.se/data/.

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





| Core | Depth (mbsf) | Length (m) | Latitude | Longitude |
|---|---|---|---|---|
| 20-GC1 | 115 | 0.83 | 77° 21.5' N | 163° 2.0' E |
| 22-PC1 | 364 | 6.49 | 78° 13.4' N | 164° 27.7' E |
| 23-GC1 | 508 | 4.06 | 78° 39.7' N | 165° 0.9' E |
| 24-GC1 | 964 | 4.05 | 78° 47.8' N | 165° 22.0' E |

**Table 1:** Location and length of sediment cores discussed in this paper.

| Sample (core, section, interval) | Mid depth (mbsf) | Lab ID | Material | $^{14}$C age (yrs BP) | Error | Cal. 2σ max age (cal yrs BP) | Cal. 2σ min age (cal yrs BP) | Median cal. age (cal yrs BP) |
|---|---|---|---|---|---|---|---|---|
| 20-GC1, CC, 2-4 cm | 0.56 | LuS11284 | Mixed benthic foraminifera: *Elphidium* spp., *Pyrgo* sp., *Islandiella teretis* | 10725 | 65 | 12511 | 11468 | 12044 |
| 20-GC1, CC, 18-20 cm | 0.72 | NOSAMS131224 | Mollusc | 11050 | 30 | 12720 | 12163 | 12521 |
| 20-GC1, CC, 20-22 cm | 0.74 | LuS11285 | Mollusc: *Macoma* sp. | 10110 | 55 | 11263 | 10715 | 11034 |
| 20-GC1, CC, 22-24 cm | 0.76 | NOSAMS131225 | Mollusc | 10050 | 40 | 11200 | 10693 | 10968 |
| 20-GC1, CC, 27-29 cm | 0.81 | LuS11286 | Mollusc: *Macoma* sp. | 11785 | 65 | 13439 | 12929 | 13209 |
| 20-GC1, CC, 27-29 cm | 0.81 | NOSAMS131226 | Mollusc | 10900 | 60 | 12619 | 11953 | 12318 |
| 23-GC1, 2, 62-79 cm | 1.77 | Lu131228 | Mollusc | 33200 | 560 | 38500 | 35700 | 37000 |
| 24-GC1, 2, 87-89 cm | 1.92 | Lu131229 | Planktonic Foraminifera, *N. pachyderma* | 43000 | 1800 | 49800 | 43700 | 46300 |

5 **Table 2:** Radiocarbon dates and calibrations from sediment cores 20-GC1, 23-GC1 and 24-GC1. All dates were calibrated with the Marine13 calibration curve (Reimer et al 2013), with ΔR = 50 ± 100 years for 20-GC1 (Cronin et al, this issue) and ΔR = 0 years for 23-GC1 and 24-GC1.



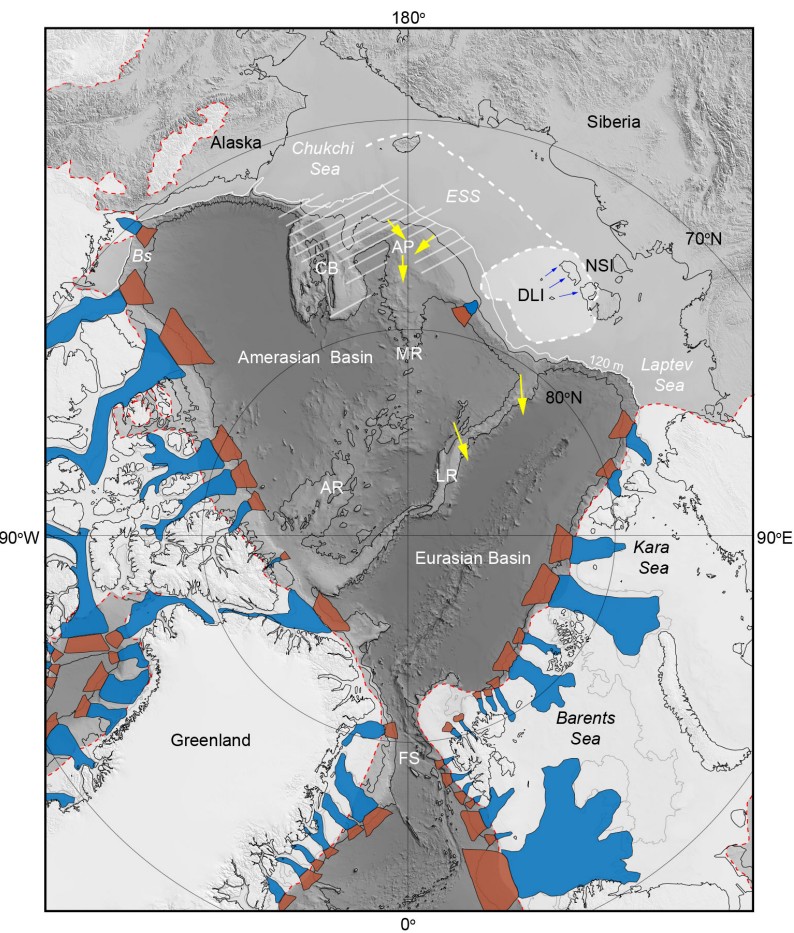

**Figure 1:** Map of the Arctic Ocean showing the maximum extent of Quaternary glaciations (red dashed line)
(Jakobsson et al., 2014). The 120 m isobath is highlighted across the Siberian, Chukchi and Beaufort Seas to
highlight the potential extent of exposed land during the global eustatic low stand of the LGM. Yellow arrows
indicate the direction of ice flow inferred from the orientation of glacial landforms on the Deep Arctic seafloor
(Jakobsson et al., 2016). Glacial extents, and flow directions, around the New Siberian Islands (NSI) and De
Long Islands (DLI) are redrawn from Basilyan et al., (2008). Dashed lines across the East Siberian shelf and
hatching on the Chukchi Borderland (CB) indicate areas of probable glacial ice in the late Quaternary (Jakobsson



et al., 2014). Known glacially excavated cross shelf troughs (blue) and trough mouth fans (brown) are redrawn from Batchelor and Dowdeswell (2014) – the exception is the single trough on the East Siberian shelf, which is described in this manuscript. AP – Arliss Plateau, AR – Alpha Ridge, BS – Beaufort Sea, FS – Fram Strait, MR – Mendeleev Ridge, LR – Lomonosov Ridge.

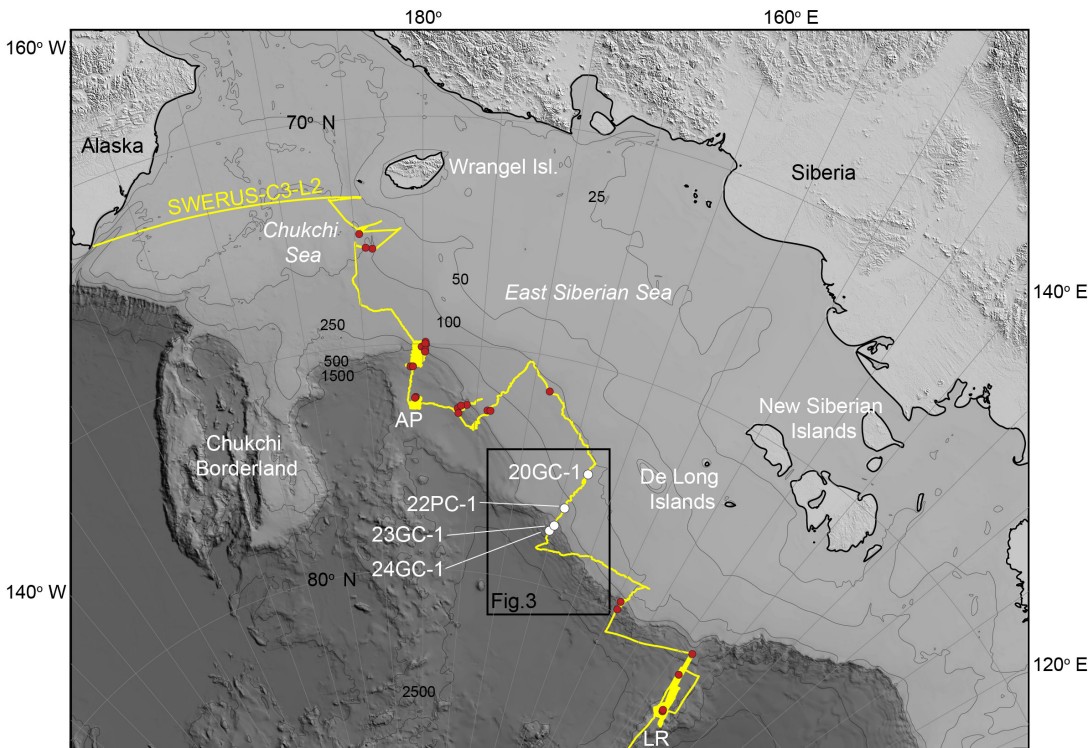

**Figure 2:** Ship track and coring sites during SWERUS-C3-L2. Location of the geophysical data and sediment cores discussed in this paper is highlighted.




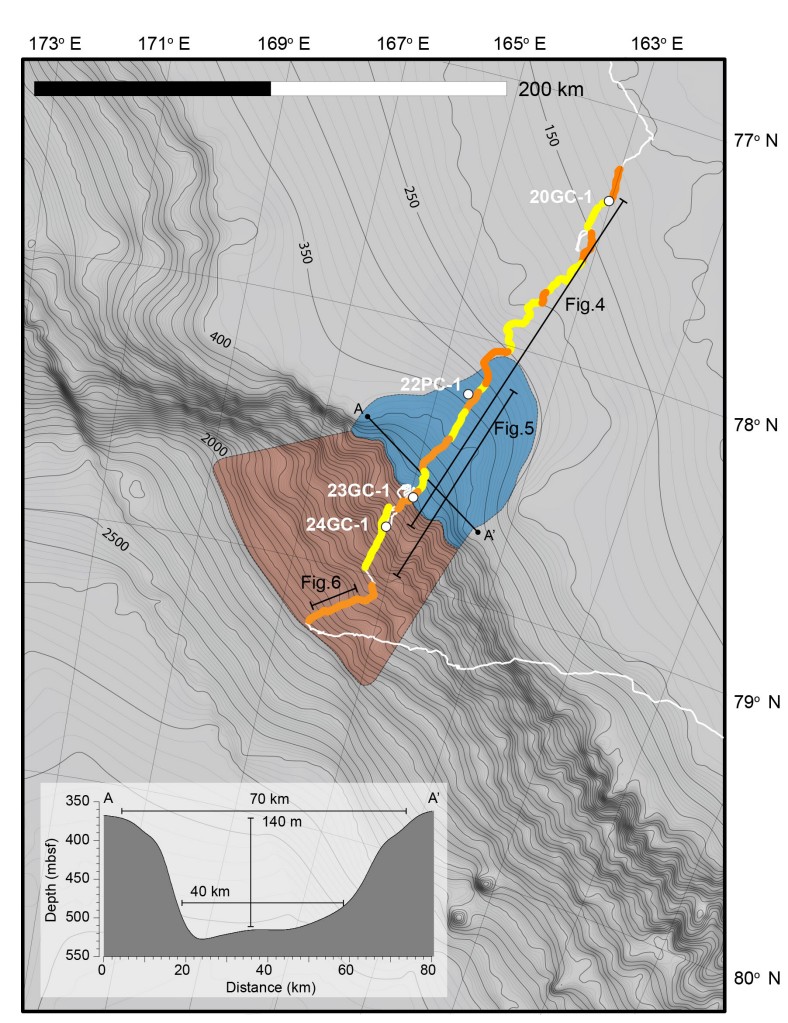

**Figure 3:** Detail of study area showing ship track with the yellow and orange lines indicating portions of the CHIRP sub-bottom data presented in Figures 4 and 5. Locations of sediment cores discussed in text are shown. Blue and brown shading represents the bounds of the glacial trough and trough mouth fan deposits (respectively)



as interpreted from the geophysical data collected during SWERUS-C3-L2. Bathymetry from IBCAO (Jakobsson et al., 2012), with 50 m major contour intervals.

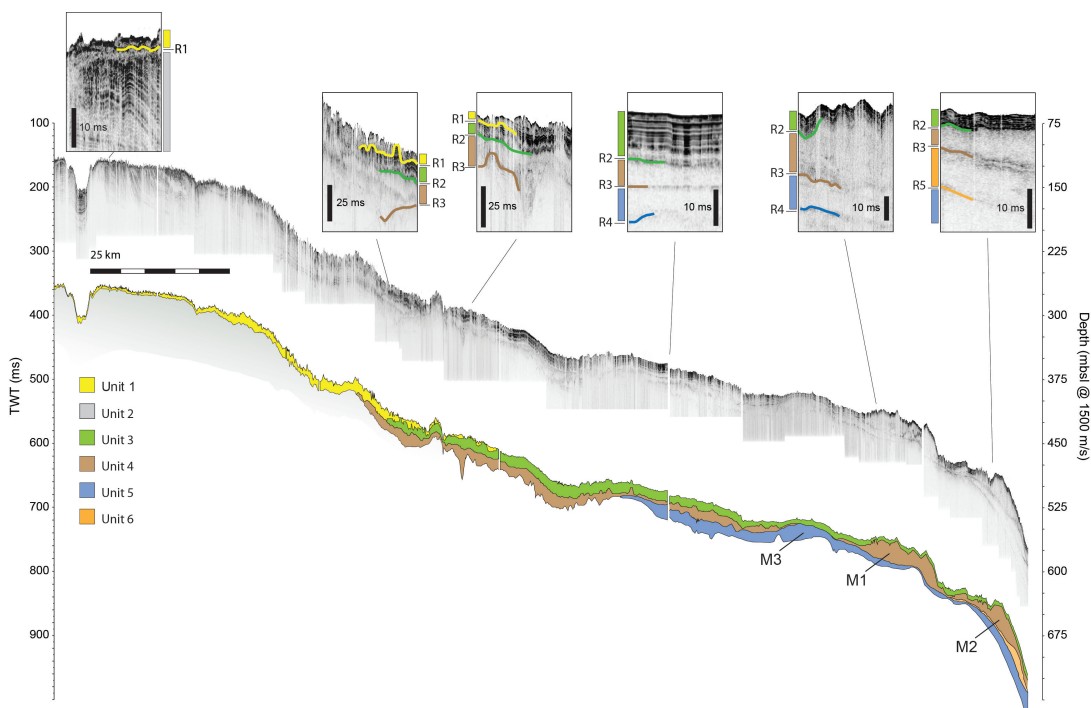

**Figure 4:** Composite CHIRP sub-bottom profile from the shallow East Siberian shelf to the shelf break at ∼ 500 mbsl. Depths are interpreted using a constant seawater velocity of 1500 m/s. An interpreted profile with six acoustic units is shown below the sub-bottom data, it is offset by 200 ms TWT from the true depth. Details of the acoustic units and traced reflectors are shown in detailed images. Acoustic Unit 1 transitions into an acoustically laminated and undisturbed drape of sediment (Unit 3) below ∼260 mbsl. Location of profile is shown in Figure 3.




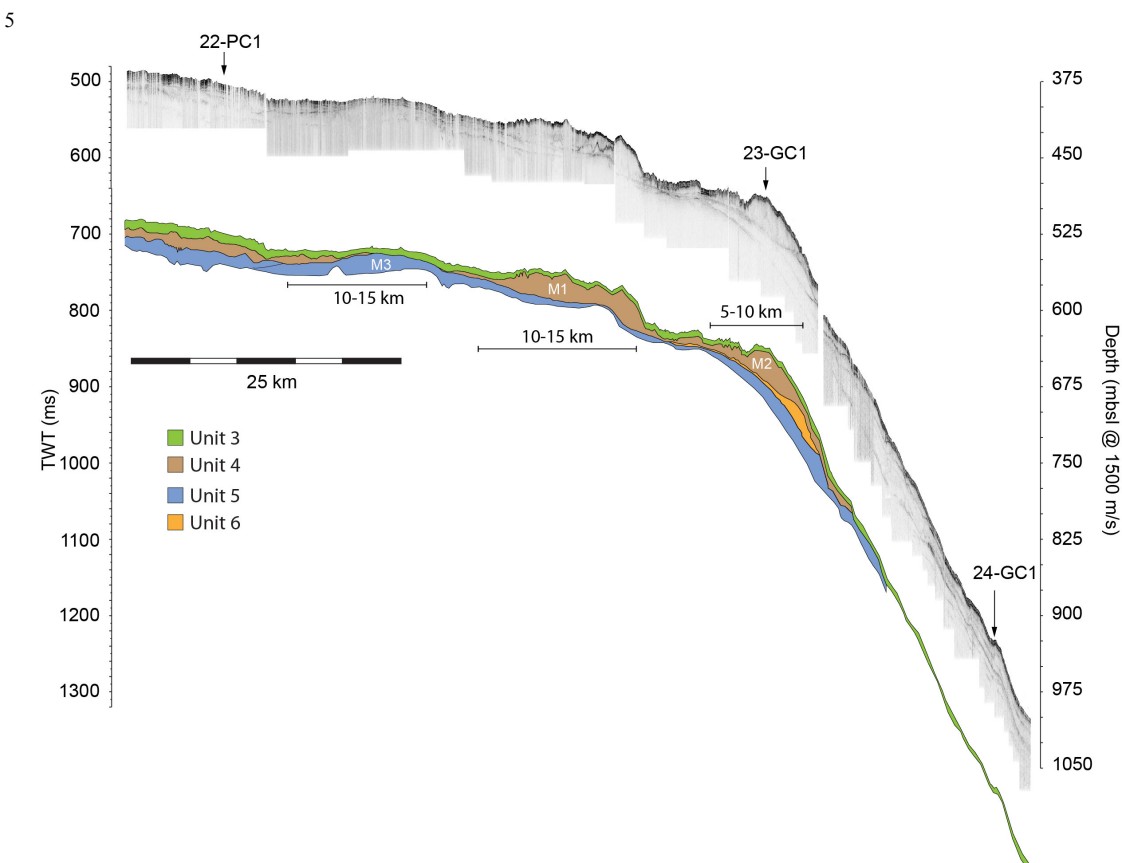

**Figure 5:** Location of sediment cores 22-PC1, 23-GC1 and 24-GC1 along the composite CHIRP sub-bottom profile. Interpreted profile is offset by 200 ms TWT from the true depth. Location of profile is shown in Figure 3.



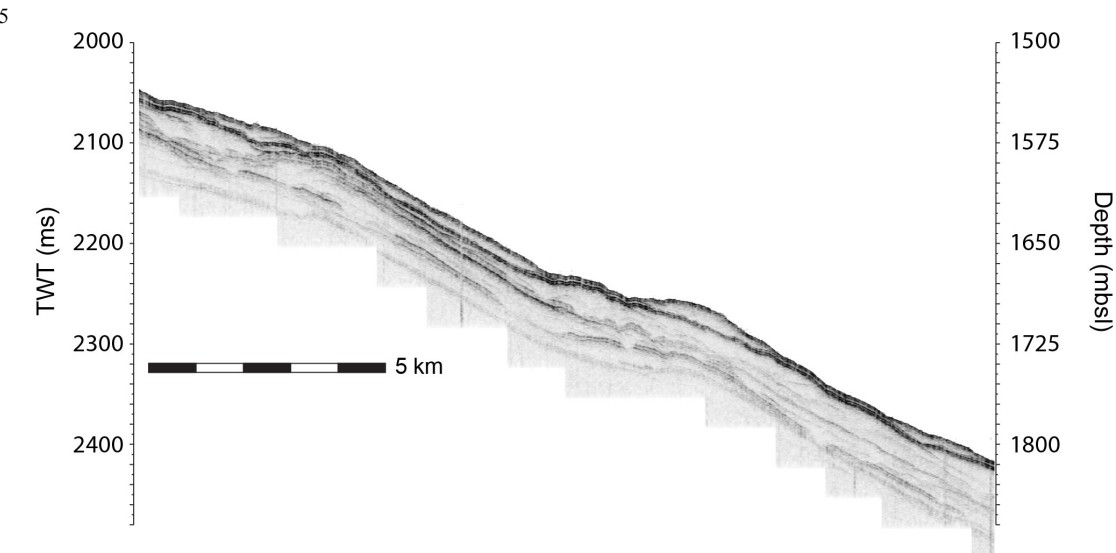

**Figure 6:** Representative acoustic stratigraphy of the continental slope, where a thin acoustically laminated veneer of sediments overlies a series of stacked acoustically transparent intervals. Some of the acoustically transparent intervals are laterally continuous, but all exhibit substantial downslope variations in thickness. Location of profile is shown in Figure 3.





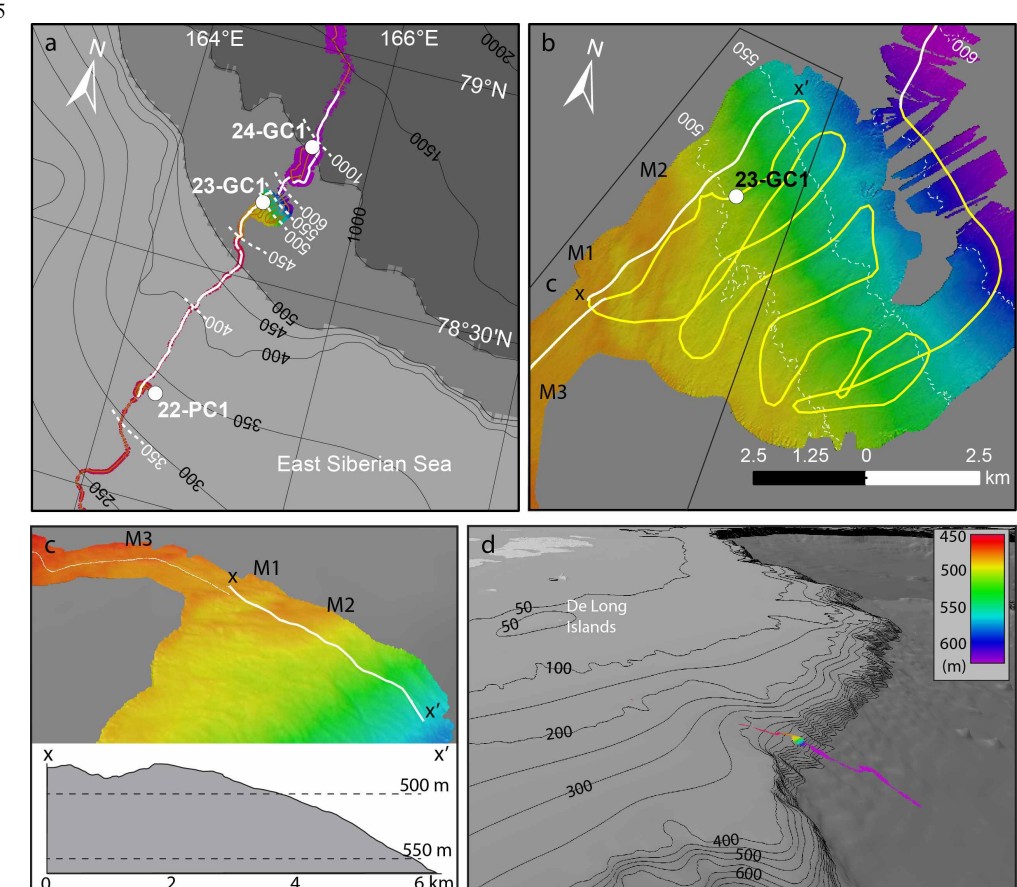

**Figure 7:** Multibeam mapping results. **A)** Ship track and extent of mapping in the study area. Measured depths (white numbers and dashed lines) are shown in comparison with gridded depths in IBCAO. **B)** Surveyed region on the outer shelf where the sedimentary mounds (grounding zone deposits) (M1, M2, and M3) are identified in the sub-bottom data. **C)** Oblique view and bathymetric cross-section of the interpreted grounding zone deposits **D)** Oblique view of the outer East Siberian shelf with the De Long Islands and New Siberian Islands in the upper left. Multibeam data is overlain on IBCAO.





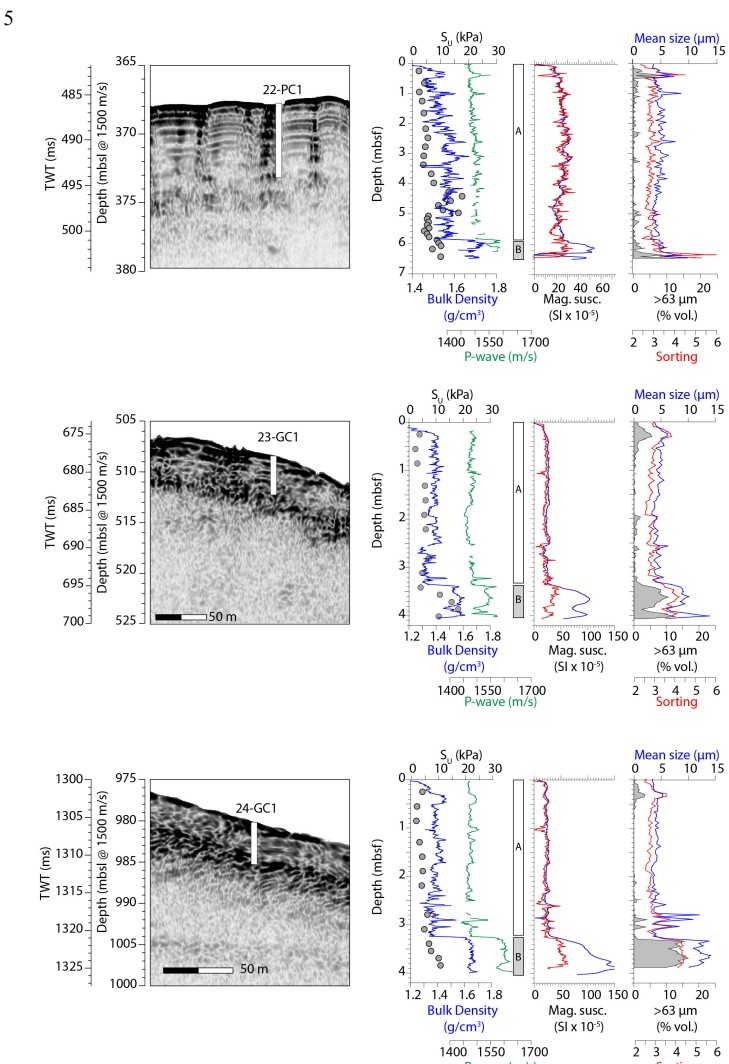

**Figure 8:** Detailed view of sub-bottom data, penetration and physical property measurements of cores 22-PC1, 23-GC1 and 24-GC1. All cores penetrate down to a strong reflector that marks the top of an





acoustically transparent interval. The base of all the cores recovered a coarser-grained, poorly-sorted unit displaying a higher bulk density and magnetic susceptibility. This is interpreted as a diamict.

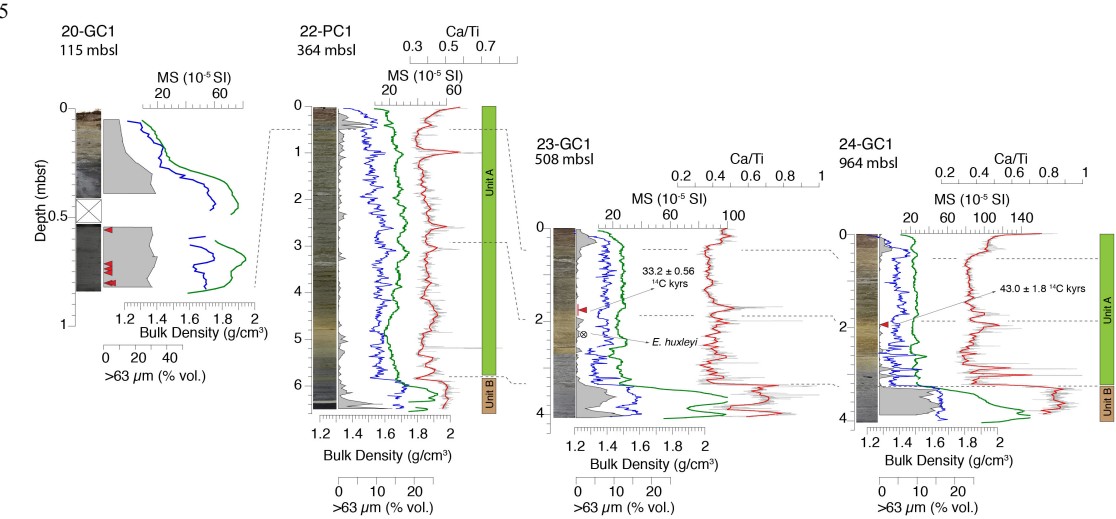

**Figure 9:** Stratigraphic correlation of core 20-GC1, 22-PC1, 23-GC1 and 24-GC1, based on MSCL, XRF-scanning, digital images and radiocarbon dating results. The acoustically laminated sediments of Unit 3 (Figure 4 and 5) are represented by sedimentary Unit A in cores 22-PC1, 23-GC1 and 24-GC1. These undisturbed sediments overly the glacial diamict of Unit B, which corresponds to acoustic Unit 4. The base of Unit A is older than ~ 50 cal. kyrs BP based on results from radiocarbon dating. Core 20-GC1, was obtained from 115 mbsl. It was taken from an area of the shelf that would have been exposed during the last glacial cycle. Radiocarbon dates from below 55 cm in the core indicate a deglacial age for the lowermost sediments. These likely correlate to the slightly coarser grained interval seen in the upper few decimeters of cores 22-PC1, 23-GC1 and 24-GC1.