# Peer review of "De Long Trough: A newly discovered glacial trough on the East Siberian Continental Margin"

_Climate of the Past, 2017_

## Referee Comment (RC1) · Anonymous Referee #1 · 12 Jun 2017

The Paper "De Long Trough: A newly discovered glacial trough on the East Siberian Continental Margin" addresses the existence of cross-shelf trough(CST) in the East Siberian Sea (ESS). Many CSTs are found on the Beaufort Sea, off Greenland, and Barents-Kara Sea of the Arctic Ocean, however, no direct evidence of CTS has been found on the ESS yet. This makes it difficult to understand glacial history in the ESS, even though many studies provided several proofs of Pleistocene ice sheet. Results from this paper are expected to contribute to reveal the existence of the fast-streaming ice sheet as well as the glacial history in the ESS.

In this paper, several images of SBP related to grounding-zone wedges(GZW) are presented. Because GZWs are generally found in CST or major fjord system, there is no doubt of the the existence of fast-streaming ice sheet in the De Long Trough.

[Figure]

However, some evidences supporting the existence of CST need to be examined carefully. 1. One of evidences for CST is a topographical depression in the IBCAO chart. The IBCAO version 3.0 is known to be compiled using Digital Bathymetric Model (DBM) and collected bathymetric data. In the ESS, however, limited measured bathymetric data may give rise a limited accuracy of IBCAO data in the region. As this paper presented the maximum depth difference between field measurement and IBCAO data reached more than 100 meters. Considering this error range, the depression depth of 140 meter measured using IBCAO data may have another geologic interpretation than CST.

2. In this paper, trough mouth fan (TMF) was presented to support CST. The only supporting evidence of TMF is a topographical feature from IBCAO data. Like the previous reason of poor accuracy of IBCAO data in the ESS, it is not easy to accept as a supporting evidence of TMF.

---

## Referee Comment (RC2) · Anonymous Referee #2 · 13 Jun 2017

The paper 'De Long Trough: A newly discovered glacial trough on the East Siberian Continental Margin' presents new geophysical and geological data from the East Siberian outer shelf and slope. Very little is presently known about the glacial history and past ice-sheet dynamics of this region, making this study particularly relevant to ideas about a Siberian Ice Sheet and an Arctic ice shelf. I have a few comments about the structure of the paper and the interpretations that are presented.

Major comments

- There could be further separation of description and interpretation in the manuscript. At present, the interpretations occur in the Discussion section and it is not clear what each of the six acoustic units have been interpreted as. Perhaps include separate description and interpretation headers within each of the results sub-chapters.

- Some of the architectural features and landforms that are described in the Discussion should be introduced earlier in the manuscript. For example, the first paragraph of section 4.1 is background information that would better fit in an introduction. The introduction could include a short paragraph on the landforms that are typically associated with ice streams, e.g. GZWs, TMFs and mega-scale glacial lineations. This section would also benefit from additional references about GZWs, TMFs and moraines.

- There is some discussion of the atypical characteristics of the De Long trough, i.e. it is quite shallow compared with other cross-shelf troughs. There could be further discussion of the ways in which the De Long trough differs from other cross-shelf troughs/ the limitations of the available data, e.g. the trough doesn't appear to cross the entire shelf, e.g. the seafloor becomes deeper towards the shelf break, whereas most Arctic cross-shelf troughs have a reverse gradient.

Minor comments

- Are any iceberg ploughmarks detected on the sub-bottom profiles or bathymetry?

- Start of second sentence of 4.1. Replace 'they' with 'ice streams', otherwise it could read as though cross-shelf troughs terminate in calving fronts.

- Although some GZWs are found at the shelf break, they are more often outer-shelf to mid-shelf features, and are commonly associated with shallower and/or narrower regions of a trough.

- Section 4.1.1. The fact that the landforms occur close to the shelf break isn't evidence that they are GZW, as shelf-break moraines are common in the geological record. Why does M2 more closely resemble a series of terminal moraines? Is this due to geometry/ amplitude/ length to height ratios?

- In Section 4.1.1: 'As both types of features are found at the terminus of marine based ice streams...'. The presence of moraines doesn't indicate a fast-flowing ice stream; in fact, moraines are more commonly associated with inter-ice stream regions. Both

features do indicate the presence of grounded ice at the shelf break.

- Fig. 6. Labels could be added to this figure, e.g. acoustically transparent intervals.

- Fig. 7. The numbers in panels a and b should be rotated so they can be read. Where is the shelf break in panel a? This could be marked on, e.g. with a dashed white line. Panel a could be rotated to that the shelf is at the top of the image and the slope at the bottom, as in the other figures. Label the GZW in the profile in c.

---

## Author Comment (AC1) · 30 Jul 2017

The Paper "De Long Trough: A newly discovered glacial trough on the East Siberian Continental Margin" addresses the existence of cross-shelf trough(CST) in the East Siberian Sea (ESS). Many CSTs are found on the Beaufort Sea, off Greenland, and Barents-Kara Sea of the Arctic Ocean, however, no direct evidence of CTS has been found on the ESS yet. This makes it difficult to understand glacial history in the ESS, even though many studies provided several proofs of Pleistocene ice sheet. Results from this paper are expected to contribute to reveal the existence of the fast-streaming ice sheet as well as the glacial history in the ESS. In this paper, several images of SBP

related to grounding-zone wedges (GZW) are presented. Because GZWs are generally found in CST or major fjord system, there is no doubt of the existence of fast-streaming ice sheet in the De Long Trough.

However, some evidences supporting the existence of CST need to be examined carefully.

1. One of evidences for CST is a topographical depression in the IBCAO chart. The IBCAO version 3.0 is known to be compiled using Digital Bathymetric Model (DBM) and collected bathymetric data. In the ESS, however, limited measured bathymetric data may give rise a limited accuracy of IBCAO data in the region. As this paper presented the maximum depth difference between field measurement and IBCAO data reached more than 100 meters. Considering this error range, the depression depth of 140 meter measured using IBCAO data may have another geologic interpretation than CST.

"We appreciate the concerns of reviewer #1 in this regard, and are acutely aware of the uncertainties in IBCAO. The manuscript is built on data collected during SWERUS-C3. The evidence for glacial activity extending out to the shelf edge is from our mapping and subbottom data.

We recognise that IBCAO Version 3.0 is, in this area, completely based on digitized contours from the Russian bathymetric map published by the Head Department of Navigation and Oceanography (HDNO) in 1999 (Naryshkin, 1999, Naryshin 2001). The source data of the Russian HDNO maps are unfortunately not publicly available. However, since the maps were compiled by a "neutral bathymetrist" and not by glacial geologist, we have no reason to believe that a bathymetric trough, here called the De Long Trough, at this location is "over interpreted" from the source data.

We have added a similar paragraph to the discussion section of the manuscript. However, in addition to this it is critical to remember that our interpretation of the broad topographic features seen in IBCAO is based on our more detailed but spatially limited mapping conducted during SWERUS-C3. "

[Figure]

2. In this paper, trough mouth fan (TMF) was presented to support CST. The only supporting evidence of TMF is a topographical feature from IBCAO data. Like the previous reason of poor accuracy of IBCAO data in the ESS, it is not easy to accept as a supporting evidence of TMF.

"In the manuscript we also present and interpret subbottom data that very convincingly shows a sequence of stacked mass-wasting (glaciogenic debris flow) deposits in front of De Long Trough. These coincide with a notable bulge in the slope profile seen in IBCAO. Together these are argued to be evidence for a TMF.

Further evidence for a glaciogenic origin for these deposits, is the dating of the uppermost acoustically laminated unit, whose base must be older than 45-50 ka (based on the radiocarbon dating in cores 23-GC and 24-GC. This is significant because it means that the mass-wasting deposits are not generated in response to transgression/regression during normal glacial cycles. If they were, the uppermost acoustically laminated unit on the slope would have been disrupted by mass-wasting during the last glacial maximum. "

---

## Author Comment (AC2) · 30 Jul 2017

The paper 'De Long Trough: A newly discovered glacial trough on the East Siberian Continental Margin' presents new geophysical and geological data from the East Siberian outer shelf and slope. Very little is presently known about the glacial history and past ice-sheet dynamics of this region, making this study particularly relevant to ideas about a Siberian Ice Sheet and an Arctic ice shelf. I have a few comments about the structure of the paper and the interpretations that are presented.

"In general we have revised the structure of the manuscript in accordance with the reviewers comments. How this was done is outlined below following each comment"

[Figure]

Major comments - There could be further separation of description and interpretation in the manuscript. At present, the interpretations occur in the Discussion section and it is not clear what each of the six acoustic units have been interpreted as. Perhaps include separate description and interpretation headers within each of the results sub-chapters.

"We added a separate subsection at the end of the results section that provides an interpretation of the acoustic stratigraphy. This interpretation is based on the combined subbottom, bathymetry and coring data – and because of this we feel it fits better after each of these data-sets are described."

- Some of the architectural features and landforms that are described in the Discussion should be introduced earlier in the manuscript. For example, the first paragraph of section 4.1 is background information that would better fit in an introduction. The introduction could include a short paragraph on the landforms that are typically associated with ice streams, e.g. GZWs, TMFs and mega-scale glacial lineations. This section would also benefit from additional references about GZWs, TMFs and moraines.

"We have moved the first paragraph of the Discussion into the introduction, and included a short paragraph on CSTs, MSGLs, GZWs, and TMFs. It includes more detailed referencing, but admittedly makes use of some of the most recent and extensive review papers of high latitude cross-shelf troughs and grounding zone wedges."

- There is some discussion of the atypical characteristics of the De Long trough, i.e. it is quite shallow compared with other cross-shelf troughs. There could be further discussion of the ways in which the De Long trough differs from other cross-shelf troughs/ the limitations of the available data, e.g. the trough doesn't appear to cross the entire shelf, e.g. the seafloor becomes deeper towards the shelf break, whereas most Arctic cross-shelf troughs have a reverse gradient.

"Many of the smaller high Arctic cross-shelf troughs do not display a reverse gradient, which tends to be a feature of some of the larger Antarctic troughs and those found

around Southern Greenland. We have added this to the discussion section. We also highlight (Figure 3) that the landward extent of the trough is uncertain. Our interpretation of its extent is anchored in the data we have collected. Without more detailed bathymetric data, it is difficult to establish its exact origin."

Minor comments - Are any iceberg ploughmarks detected on the sub-bottom profiles or bathymetry? "Yes, iceberg ploughmarks are very common above water depths of 280-320 mbsl. This has been added to the description of the Bathymetry data."

- Start of second sentence of 4.1. Replace 'they' with 'ice streams', otherwise it could read as though cross-shelf troughs terminate in calving fronts. "This has been corrected."

- Although some GZWs are found at the shelf break, they are more often outer-shelf to mid-shelf features, and are commonly associated with shallower and/or narrower regions of a trough. "A more detailed, but concise description of GZW location has been included in the introduction."

- Section 4.1.1. The fact that the landforms occur close to the shelf break isn't evidence that they are GZW, as shelf-break moraines are common in the geological record. Why does M2 more closely resemble a series of terminal moraines? Is this due to geometry/ amplitude/ length to height ratios?

"Our initial interpretation was based on their geometry, namely that M2 shows pronounced lateral variations in thickness. In the revised manuscript we have included estimates of their length to height ratios (165:1 to 600:1), which are clearly more similar to those reported for GZWs compared to terminal moraines (<10:1). We have elaborated on these observations in subsection '4.1.2 Grounding line deposits'. However, we still believe that the existing mapping data is too limited to unequivocally define these glacial landforms."

- In Section 4.1.1: 'As both types of features are found at the terminus of marine based

ice streams. . . '. The presence of moraines doesn't indicate a fast-flowing ice stream; in fact, moraines are more commonly associated with inter-ice stream regions. Both features do indicate the presence of grounded ice at the shelf break. "This is true, and we have corrected the text accordingly. We cannot unequivocally determine if the features are terminal moraines or GZWs, and recognize that the different interpretations are critical for interpreting the ice dynamics. We have made this clearer in the text."

- Fig. 6. Labels could be added to this figure, e.g. acoustically transparent intervals. "The figure has been annotated to better illustrate the different units."

- Fig. 7. The numbers in panels a and b should be rotated so they can be read. Where is the shelf break in panel a? This could be marked on, e.g. with a dashed white line. Panel a could be rotated to that the shelf is at the top of the image and the slope at the bottom, as in the other figures. Label the GZW in the profile in c. "These changes have been made."
* * *
[Figure]

**Fig. 1.** Figure 6 edited after review

**a**

164°E 166°E

79°N

**24-GC1**

**23-GC1**

1000

600

550

500

500

450

**22-PC1** 350

shelf break

East Siberian Sea

2000

1500

1000

500

450

400

400

350

300

250

**b**

N

550

600

500

M2

**23-GC1**

x'

M1

c x

M3

2.5   1.25   0       2.5
km

**c**

M3

M1

x

M2

x'

x M1        M2              x'

500 m

550 m

0        2        4      6 km

**d**

De Long
Islands

50

50

100

200

300

400
500
600

450
500
550
600
(m)

**Fig. 2.** Figure 7 edited after review